# A Preliminary Investigation of Interspecific Chemosensory Communication of Emotions: Can Humans (*Homo sapiens*) Recognise Fear- and Non-Fear Body Odour from Horses (*Equus ferus caballus*)

**DOI:** 10.3390/ani11123499

**Published:** 2021-12-08

**Authors:** Agnieszka Sabiniewicz, Michał Białek, Karolina Tarnowska, Robert Świątek, Małgorzata Dobrowolska, Piotr Sorokowski

**Affiliations:** 1Institute of Psychology, University of Wrocław, 50-527 Wrocław, Poland; michal.bialek3@uwr.edu.pl (M.B.); k.a.c.tarnowska@gmail.com (K.T.); mrswiatek@wp.pl (R.Ś.); sorokowskipiotr@yahoo.co.uk (P.S.); 2Department of Otorhinolaryngology, Smell and Taste Clinic TU Dresden, 01307 Dresden, Germany; 3International Center for Interdisciplinary Research, Silesian University of Technology, 44-100 Gliwice, Poland; Malgorzata.Dobrowolska@polsl.pl

**Keywords:** interspecific communication, chemosignals, emotional recognition, fear, horses, horse–human bond

## Abstract

**Simple Summary:**

Thus far, little attention has been paid to interspecific odour communication of emotions, and no studies have examined whether humans are able to recognise animal emotions from body odour. Thus, the aim of the present study was to address this question. Body odour samples were collected from 16 two-year-old thoroughbred horses in fear and non-fear situations, respectively. The horse odour samples were then assessed by 73 human odour raters. We found that humans, as a group, were able to correctly assign whether horse odour samples were collected under a fear- or a non-fear condition, respectively. An open question remains, which is whether humans could simply distinguish between little versus much sweat and between high intensity versus low intensity or were able to recognise horses’ fear and non-fear emotions. To conclude, the present results indicate that olfaction might contribute to the human recognition of horse emotions. However, these results should be addressed with caution in light of the study’s limitations and only viewed as exploratory for future studies.

**Abstract:**

Mammalian body odour conveys cues about an individual’s emotional state that can be recognised by conspecifics. Thus far, little attention has been paid to interspecific odour communication of emotions, and no studies have examined whether humans are able to recognise animal emotions from body odour. Thus, the aim of the present study was to address this question. Body odour samples were collected from 16 two-year-old thoroughbred horses in fear and non-fear situations, respectively. The horse odour samples were then assessed by 73 human odour raters. We found that humans, as a group, were able to correctly assign whether horse odour samples were collected under a fear- or a non-fear condition, respectively. Furthermore, they perceived the body odour of horses collected under the fear condition as more intense, compared with the non-fear condition. An open question remains, which is whether humans could simply distinguish between little versus much sweat and between high intensity versus low intensity or were able to recognise horses’ fear and non-fear emotions. These results appear to fit the notion that the ability to recognise emotions in other species may present an advantage to both the sender and the receiver of emotional cues, particularly in the interaction between humans and domesticated animals. To conclude, the present results indicate that olfaction might contribute to the human recognition of horse emotions. However, these results should be addressed with caution in light of the study’s limitations and only viewed as exploratory for future studies.

## 1. Introduction

Olfactory communication via chemosignals is one of the most common ways in which animals convey information between conspecifics [1]. The natural body odour of animals and humans consists of a wide range of volatile compounds [2] that carry a variety of information. This information has been shown to also play an important role in human (Homo sapiens) non-verbal communication [3,4,5,6,7,8,9]. Body odour conveys cues that are essential for intraspecific social communication (for animals, see [1]; for humans, see [10]). This includes information about the emotional state of a conspecific (for non-human animals, see [11]; for humans, see [12]).

From a neuroanatomical perspective, odour information is directly linked to brain areas such as the amygdala, the hippocampus, and orbitofrontal cortex which are strongly associated with emotion processing and survival responses [13,14,15,16]. A growing number of studies suggest that the ability to recognise emotions based on odour cues emitted by conspecifics is shared by a number of social mammal species including gorillas [17], dogs [18], horses [19], and rats [20]. Animal body odour has been found to convey information about fear [21], withdrawal, and submission [22], as well as about dominance [23]. In a more recent study, Krueger and Flauger [19] found that horses spent most time sniffing the faeces of conspecifics from which they received the highest amount of aggressive behaviour. 

Some of these findings are also consistent with reports in humans [8], as human body odour, too, carries information about one’s emotional state [5,6,7,12,24,25,26,27]. Humans are able to recognise emotions such as happiness [12,28,29] or fear [27,29,30] from the body odour of conspecifics. 

The ability to recognise and respond appropriately to volatile chemicals carrying information about the emotional state of a conspecific is thought to provide benefits for both the signaller and the receiver. The processing of chemosensory anxiety signals affects perceptional performance in humans by increasing cognitive alertness [31]. The ability to recognise a conspecific’s emotions appears to be crucial for appropriate social interaction, particularly for group organisation [32], which is essential in social species such as, for example, horses [33].

Considering the intimate and long-lasting relationship between human subjects and domesticated animals, the question that arises is whether the ability to recognise emotions conveyed by chemosignals exists only within species or can also occur between species. For example, the domestication of horses started about 6000 years ago, and since then, they have been used for transport, herding, food, trade, welfare, competition, or recreation [34,35]. Nowadays, horses are commonly used as companion animals [36] or in therapeutic riding programmes [37,38,39]. All these are examples of interspecific relationships and suggest a possibly important role of the communication of emotions between humans and animals.

Interspecific odour communication of emotions has received increased attention in recent years [19,40,41,42,43,44]. D’Aniello et al. [41] reported that dogs are able to recognise human emotions from body odour, as they displayed behaviours indicating stress only when being presented with human odour of fear. In a recent study, Sabiniewicz et al. [44] demonstrated that horses presented differential behaviours in response to human fear and non-fear odour samples, showing the ability of purely olfactory recognition of human emotions. Moreover, human body odours were shown to induce systematic sympathetic and parasympathetic changes in horses, revealing that the human chemosignals are able to induce similar emotional status in horses [43,45]. In contrast, no studies so far have examined whether humans are able to recognise animal emotions from body odour. Thus, the aim of the present study was to address this question. We decided to use horses as odour donors, due to their long history of close coexistence with humans.

## 2. Materials and Methods

### 2.1. Odour Sampling

Odour sampling was conducted at Wrocław Horse Racing Stable Partynice in December 2018. Odour donors were 16 two-year-old thoroughbred horses (*Equus ferus caballus*): 11 males and 5 females. In the first step, pieces of near-odourless fleece material (30 × 50 cm, Jysk) were placed under new saddle pads. Then, the horses took part in a 20 min racing training. Since it was one of their first training, we expected that the horses might experience fear due to the stress related to carrying a rider and taking part in a race. These events are commonly known to be particularly stressful for inexperienced horses [46]. Both ears position, head position, and body tension of the horses during this training confirmed our expectation. Immediately after the training, the pieces of fleece material were retrieved, put into airtight sealed bags, marked for identification, and placed in a −20 °C freezer. In the second step, after a minimum of 20 min, the horses were cleaned with fresh straw and brushed, and new pieces of fleece material of the same dimensions, together with new saddle pads and lunging belts, were put on them. Then, they were brought to a horse training carousel, where they walked for the next 30 min, relaxing after the effort of the race. After that episode, again, the pieces of fleece material were retrieved, put into airtight sealed bags, marked for identification, and placed in a freezer. Since the horses were familiar with the training carousel, we expected that they should not experience fear during this episode. Here, too, ear position, head position, body tension, and behaviour of the horses confirmed our expectation.

### 2.2. Odour Rating

The odour rating occurred in a quiet, ventilated room in February 2019. The horse odour samples (defrosted pieces of fleece material) were cut into four equally sized pieces, one of which was used and assessed individually by 73 odour raters (49 women and 24 men) aged 18–35 years (mean = 21.9, standard deviation (SD) = 3.66). Each of 16 pairs of odours contained a sample collected during the fear condition and a sample collected from the same horse during the non-fear condition, placed in two separate closed glass jars of 0.5 litres. The odour rater was allowed to briefly sniff at both samples for a maximum of two inhalations and was then asked to assess which of the two jars contained the fear odour and which one contained the non-fear odour. Each response was scored as either correct or incorrect. The odour rater was also asked to assess whether the two samples of a given odour pair were equally intense or if one was more intense than the other–in the latter case, which one. After the odour rating session, each participant completed a short questionnaire concerning health state to exclude the presence of acute or chronic olfactory impairment [47] and experience he or she had had with horses. Specifically, we asked participants about their experience with horses in terms of horse riding (‘yes’–‘no’; if yes, for how long) and owning a horse. 

### 2.3. Ethical Note

This study was approved by the Institutional Review Board of the University of Wroclaw (Protocol Number 2018/JPE0015). All procedures performed in this study involving human participants were in accordance with the 1964 Helsinki declaration and its later amendments or comparable ethical standards. Informed consent was obtained from the human odour raters and from the horse trainers of all horses included in the study. All procedures performed in this study involving animals were in accordance with the ethical standards of the institution or practice at which the studies were conducted. 

### 2.4. Data Analysis

As the raters provided 16 binary decisions, we divided the number of correct classifications by the total number of assessments, thus obtaining the accuracy score. This accuracy score can be treated as a continuous variable, because of the operationalisation, and because the underlying psychological contract in its nature is continuous, rather than categorical. That is, people recognise smell with some accuracy (continuous variable), which, in turn, translates into the probability of correct classification in each trial (categorical variable).

The accuracy score for all human raters as a group was analysed by means of a one-sample *t*-test against the guessing accuracy of 50%. If significant. this analysis would show that, in general, raters classify smell as fear or non-fear with accuracy above guessing. We followed up this analysis with binomial tests on an individual level, estimating the number of above-chance correct raters. 

These analyses, however, inform us about who the best classifiers are, and which conditions facilitate correct smell recognition. To this end, we considered possible moderators of the accuracy of the smell classification: (1) male and female raters with regard to their number of correct responses; (2) raters with and without experience with horses. These moderators were assessed using 2 × 2 between-subject Analysis of Variance (ANOVA) with classical *p*-values and Bayesian statistics. Possible differences between the intensity of the samples were assessed by means of independent samples *t*-test. Data are presented as mean values (±standard deviation). Statistical analyses were performed using JASP v. 0.11.1 (www.jasp-stats.org, Amsterdam, The Netherlands, accessed on 1 October 2019), with *p* < 0.05 set as the level of significance. The effect sizes are accompanied by their 95% confidence intervals.

## 3. Results

Figure 1 presents the distribution of accuracy of odour sample classifications. The average accuracy was 65.7% (SD = 13.5%, range 31.3%−93.8%) and was statistically greater than chance (i.e., 50%) (*t*(72) = 9.91, *p* < 0.001, d = 1.16, 95% confidence interval (CI) (0.91–∞)). Moreover, at the individual level, 11 out of 73 subjects (4 men and 7 women) performed significantly above chance (binominal test, *p* < 0.05) in the odour rating task, and none performed significantly below chance. Therefore, we provide evidence that human individuals are able to correctly categorise horses’ fear vs. non-fear odours.

We then explored our dataset to search for moderators of the accuracy of the odour classification. To this end, we submitted our data to a 2 (experience with horses: yes, no) × 2 (gender of participant) between-subject ANOVA. None of the moderators affected the accuracy; all *p*’s > 0.375. We quantified the evidence against the effects using Bayesian statistics with default prior *r* = 0.5. The null model was indistinguishable from the model with the main effect of experience (B01 = 1.74), and from the model with sex (B01 = 2.78), but more supported by the evidence than a model with both of the main effects (B01 = 5.42). Therefore, we cannot reject the hypothesis that there are differences in accuracy caused by experience with horses, or by the sex of the participant.

Finally, we compared which odour (fear vs. non-fear) was perceived as more intense. We found that the fear odour was perceived as more intense in 64% of trials, and the non-fear odour in only 16.1% of trials and thus in a significantly higher proportion, (*t*(72) = 16.15, *p* < 0.001, d = 1.89, 95%CI (1.53–2.27)).

## 4. Discussion

The results of the present study show that humans, as a group, were able to correctly assign whether horse odour samples were collected under a fear- or a non-fear condition, respectively. Furthermore, they perceived the body odour of horses collected under the fear condition as more intense, compared with the non-fear condition.

The human sense of smell has been demonstrated to be surprisingly good when compared with that of other mammals [48,49]. Several studies have shown that humans even outperform dogs, mice, and rats with regard to their threshold sensitivity with certain odorants [50,51,52]. Human subjects have also been shown to use olfactory cues in social communication with conspecifics and are able to correctly extract emotional information from human body odours [53]. Human subjects lacking a functioning sense of smell have been reported to experience several difficulties related to the social dimension of life [54].

Recent studies have shown that humans cannot only correctly recognise chemosignals concerning, e.g., sex, age, identity, health status, and emotions of conspecifics but can also extract at least some olfactory information from the body odour of other species. Based on body odour cues alone, humans were able to correctly identify individuals of other mammals such as dogs [55] and gorillas [56]. Similarly, they were able to correctly assign individual mice to strains which only differed genetically in their major histocompatibility complex [57].

Humans and horses have cooperated in a variety of ways during the last 6000 years. Concomitantly, a variety of interspecific communication channels have evolved between humans and horses, among which the majority are based on visual [33], haptic, or vocal cues [58], but the smell of horses has also been shown to contribute to human reactions to horses. Turrell and Craig [59] showed that the odour of horses in a therapeutical context led to the decrease in cortisol level and to higher self-reported well-being in human patients. In the present study, we addressed the question of whether olfactory signals may contribute to the human recognition of horse emotions. In line with the studies showing that humans are able to use olfactory cues to recognise human emotions and can correctly interpret heterospecific odour signals, the present results indicate that olfaction might contribute to the human recognition of horse emotions. Humans appeared to correctly assign the emotional states to the horse odour samples but also perceived the body odour of horses collected under the fear condition as more intense, compared with the non-fear condition, thus displaying the basic ability to discriminate between the odour samples (based on intensity).

One plausible explanation of this result would be related to the advantage of being able to recognise emotions in other species. Emotions are considered as the causes, mediators, and consequences of our social interactions which are fundamental in our everyday lives [60]. Appropriate recognition of emotions is necessary to adapt in some specific social contexts. Investigations have shown that the major cause of horse–human accidents is unexpected fear reactions in the horse [61]. Thus, recognition of fear from horse body odour could have been useful during the history of cooperation with horses to prevent dangerous situations such as a fall from the horse or becoming injured by this animal. The current use of horses as companion or therapeutic animals may also support the ability to recognise their emotions. Being in a close relationship with an animal involves intimacy and emotional bond [62], and recognising an animal’s emotions by multiple sensory channels might be beneficial for this emotional attachment.

In the process of domestication, human beings shaped both anatomy and behaviour of animals by means of controlled breeding of individuals [63,64]. Since behaviour is related to emotions [65], conducting such a directed selection may also have favoured an increase in the understanding of emotions of domesticated animals.

The present study is not free from limitations. Firstly, the horses’ reactions in the fear condition were only validated via behavioural measures. Thus, it may be insufficient to conclude whether this condition actually generated fear and, if so, to what extent. Future research should include other options for verifying the emotional reaction of horses, such as measuring cortisol levels in horses’ blood. Secondly, the participants were not asked to openly attribute a quality to the odour samples but had to choose between limited options and closed responses—intense or not; fear or not. On the one hand, free odour naming is tiring and challenging [66], especially when participants are presented with 32 odour samples. Thus, by employing a forced recognition procedure [67], we aimed to make this task easier for the participants. On the other hand, it should be noted that this kind of odour assessment is bound to create a response bias. Thirdly, the fear condition contained a strong element of physical activity which is bound to increase sweat production. As a result, an open question remains, which is whether humans could simply distinguish between little versus much sweat and between high intensity versus low intensity or were able to recognise horses’ fear and non-fear emotions. While future research in this field should control for the level of physical activity in both conditions, it should be noted that experiencing fear per se leads to increased sweating [68]. Thus, we assume that the horses could sweat more because of increased physical activity as well as experiencing more fear. Eventually, several methodological issues should be considered. A relatively short time lag of 20 min between the fear and non-fear situations, limited possibilities of cleaning the horses between both conditions, and not randomised order of treatments are weaknesses of this study. This being said, it must be noted that the study was conducted in natural, non-laboratory conditions, and changing these aspects was not practically feasible.

## 5. Conclusions

To conclude, the results of the present study suggest that olfactory cues might contribute to the recognition of horse emotions by humans. An open question remains, which is whether humans could simply distinguish between little versus much sweat and between high intensity versus low intensity or were able to recognise horses’ fear and non-fear emotions. Future studies with carefully elaborated methodology should explore this issue further.

## Figures and Tables

**Figure 1 animals-11-03499-f001:**
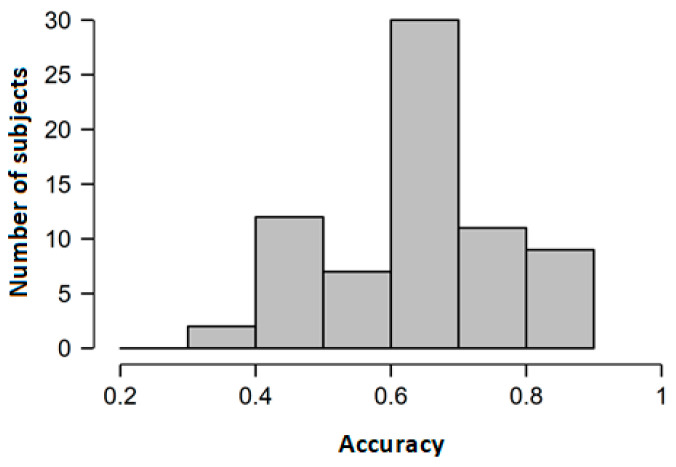
Distribution of the accuracy of odour recognition.

## Data Availability

The datasets analysed during the current study are not publicly available due to the privacy of the participants but are available from the corresponding author on reasonable request.

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
