# Peer review of "A Preliminary Investigation of Interspecific Chemosensory Communication of Emotions: Can Humans (Homo sapiens) Recognise Fear- and Non-Fear Body Odour from Horses (Equus ferus caballus)"

_animals, 2021, doi:10.3390/ani11123499_

Round 1

Reviewer 1 Report

In this paper, the authors addressed the issue of the transfer of emotion from horses to humans via chemosignals. I appreciate the effort of the authors to study the interspecies emotional transfer. The idea is new and potentially very important from an applied perspective. However, I fear that there are several serious pitfalls due to inappropriate use of methods.

First, the authors cannot be sure they are collecting fear chemosignals since stress does not generate fear in several contexts, even though the authors have collected behavioral measures.

The second problem relies on how the horses were cleaned. Can the authors be sure that fear chemosignals were completely eliminated? One of the properties of chemosignals is that they can convey information in the absence of the sender. For example, a prey can leave fear chemosignals when attacked by a predator in a place that could be useful for conspecifics after long periods.

Third and most important, the raters were made aware that fear chemosignals were in one of the two samples, which, together with the fact that the presumptive fear chemosignals were rated as more intense, biased the raters' judgments.

Reluctantly, I recommend rejecting the paper for publication.

Author Response

Dear Reviewer,

Thank you very much for expressing your positive opinion about the topic of our study. We understand your concerns regarding the manuscript itself. While suggesting major revision, other reviewers addressed the same issues. We referred to all of them describing the limitations of the study. Please find our responses below.

First, the authors cannot be sure they are collecting fear chemosignals since stress does not generate fear in several contexts, even though the authors have collected behavioral measures. The second problem relies on how the horses were cleaned. Can the authors be sure that fear chemosignals were completely eliminated? One of the properties of chemosignals is that they can convey information in the absence of the sender. For example, a prey can leave fear chemosignals when attacked by a predator in a place that could be useful for conspecifics after long periods. Third and most important, the raters were made aware that fear chemosignals were in one of the two samples, which, together with the fact that the presumptive fear chemosignals were rated as more intense, biased the raters' judgments.

Response: We carefully considered the reviewer’s suggestions and described study limitations. The corresponding paragraph now reads as follows: The present study is not free from limitations. Firstly, the horses’ reactions in the fear condition were only validated via behavioural measures. Thus, it may be insufficient to conclude whether this condition has actually generated fear and, if so, to what extent. Future research should include other options of verifying the emotional reaction of horses, such as measuring cortisol levels in horses blood. Secondly, the participants were not asked to openly attribute a quality of the odour samples but had to choose between limited options and closed responses: intense or not; fear or not. On the one hand, free odour naming is tiring and challenging [65], especially when participants are presented with 32 odour samples. Thus, by employing forced recognition procedure [66], we aimed to make this task easier for the participants. However, it should be noted that this kind of odour assessment is bound to create a response bias. Thirdly, the fear condition contained a strong element of physical activity which is bound to increase sweat production. As a result, an open question remains whether humans could simply distinguish between little versus much sweat and between high-intensity versus low-intensity or were able to recognise hoses fear and non-fear emotions. While future research in this field should control for the level of physical activity in both conditions, it should be noted that experiencing fear per se leads to increased sweating [67]. Thus, we assume that the horses could sweat more not only because of increased physical activity but also experiencing more fear. Eventually, several methodological issues should be considered. Relatively short time-lag of 20 min. between the fear and non-fear situation, limited possibilities of cleaning the horses between both conditions, and not randomised order of treatments are weaknesses of this study. This being said, it must be noted that the study was conducted in natural, non-laboratory conditions and changing these aspects was not practically feasible.

Additionally, as a response to your second comment, we described the procedure of horses’ cleaning in detail. The corresponding sentence now reads as follows: In the second step, after a minimum of 20 minutes, the horses were cleaned with fresh straw and brushed, and new pieces of fleece material of the same dimensions, together with new saddle-pads and lunging belts were put on them

Thank you very much for all your remarks. We have addressed them all. Furthermore, after having a careful consideration of your comments we decided to change the manuscript’s title into: “A preliminary investigation of interspecific chemosensory communication of emotions: humans (Homo sapiens) can recognise fear- and non-fear body odour from horses (Equus ferus caballus)”.

Please, let us underline that with all the weaknesses, this study is first to examine human ability to smell fear-and non-fear of horses via chemosignals. Thus, it can set directions for future research with a carefully planned methodology.

Reviewer 2 Report

Review interspecific communication of emotions.

I really like this topic! The paper is well written but the validity of the discussion and conclusion is questionable. There are a number of methodological problems:  

  • there is a lack of validation: the “fear-treatment” or “stress-treatment” was not systematically validated by other measures, hence no conclusions regarding fear or stress-levels can be made
  • the subjects were not asked to openly attribute a quality but had to choose between limited options and closed responses: intense or not; fear or not. This is bound to create a response bias.
  • the exercise level and amount of sweating was not controlled for (racing versus walking)
  • the order of treatments was not randomized
  • the time-lag of 20 minutes between the treatments may not be sufficient to ensure independent data
  • No other inferences can be drawn besides the conclusion that humans can distinguish between little versus much sweat and between high-intensity versus low-intensity. Future research questions and designs should be formulated that can allow for reliable and valid conclusions regarding fear levels, stress levels or emotional valence.  

The discussion needs to be rewritten accordingly. Given the severity of these methodological issues, I recommend a major revision.

Line 39: “This information has been recently shown to also play an important 39 role in human (Homo sapiens) non-verbal communication [3-9].” The references date from between 2009 to 2012. I would omit “recently”.

Line 48: put a comma after “gorillas (17)”

Line 50: “as well as about aggression and dominance [23].” The article focusses on aggression, not on dominance. Omit “dominance”.

Line 52: faces or faeces?

Author Response

Dear Reviewer,

Thank you very much for expressing your positive opinion about our manuscript. We appreciate your expert, detailed comments and the advice you gave us. Based on your suggestions, we made many changes to the manuscript. Please find our responses below.

There is a lack of validation: the “fear-treatment” or “stress-treatment” was not systematically validated by other measures, hence no conclusions regarding fear or stress-levels can be made.

esponse: We took up the reviewer’s suggestion and now address this concern. The corresponding sentences now read as follows: The present study is not free from limitations. Firstly, the horses’ reactions in the fear condition were only validated via behavioural measures. Thus, it may be insufficient to conclude whether this condition has actually generated fear and, if so, to what extent. Future research should include other options of verifying the emotional reaction of horses, such as measuring cortisol levels in horses blood.

The subjects were not asked to openly attribute a quality but had to choose between limited options and closed responses: intense or not; fear or not. This is bound to create a response bias.

Response: We carefully considered the reviewer’s suggestion and now address this concern. The corresponding sentences now read as follows: Secondly, the participants were not asked to openly attribute a quality of the odour samples but had to choose between limited options and closed responses: intense or not; fear or not. On the one hand, free odour naming is tiring and challenging [65], especially when participants are presented with 32 odour samples. Thus, by employing forced recognition procedure [66], we aimed to make this task easier for the participants. However, it should be noted that this kind of odour assessment is bound to create a response bias.

The exercise level and amount of sweating was not controlled for (racing versus walking).

Response: We took up the reviewer’s suggestion and now address this concern. The corresponding sentences now read as follows: Thirdly, the fear condition contained a strong element of physical activity which is bound to increase sweat production. As a result, an open question remains whether humans could simply distinguish between little versus much sweat and between high-intensity versus low-intensity or were able to recognise hoses fear and non-fear emotions. While future research in this field should control for the level of physical activity in both conditions, it should be noted that experiencing fear per se leads to increased sweating [67]. Thus, we assume that the horses could sweat more not only because of increased physical activity but also experiencing more fear.

The order of treatments was not randomized. The time-lag of 20 minutes between the treatments may not be sufficient to ensure independent data

Response: We carefully considered the reviewer’s suggestions and addressed these concerns. The corresponding sentences now read as follows: Eventually, several methodological issues should be considered. Relatively short time-lag of 20 min. between the fear and non-fear situation, limited possibilities of cleaning the horses between both conditions, and not randomised order of treatments are weaknesses of this study. This being said, it must be noted that the study was conducted in natural, non-laboratory conditions and changing these aspects was not practically feasible.

No other inferences can be drawn besides the conclusion that humans can distinguish between little versus much sweat and between high-intensity versus low-intensity. Future research questions and designs should be formulated that can allow for reliable and valid conclusions regarding fear levels, stress levels or emotional valence. The discussion needs to be rewritten accordingly. 

Response: We took up the reviewer’s suggestion. Except for the modifications described above, we referred with criticism to the present results in the simple summary, abstract, discussion and conclusions. The corresponding sentences now read as follows: However, these results should be addressed with caution in light of the study's limitations and only viewed as exploratory for future studies. (…). To conclude: the results of the present study suggest that olfactory cues might contribute to the recognition of horse emotions by humans. An open question remains whether humans could simply distinguish between little versus much sweat and between high-intensity versus low-intensity or were able to recognise hoses fear and non-fear emotions. Future studies with carefully elaborated methodology should explore this issue further.

Furthermore, we decided to change the manuscript’s title into: “A preliminary investigation of interspecific chemosensory communication of emotions: humans (Homo sapiens) can recognise fear- and non-fear body odour from horses (Equus ferus caballus)”.        

Line 39: “This information has been recently shown to also play an important 39 role in human (Homo sapiens) non-verbal communication [3-9].” The references date from between 2009 to 2012. I would omit “recently”.

Response: Thank you very much. We have deleted “recently”.

Line 48: put a comma after “gorillas (17)”

Response: Thank you, we have corrected it.

Line 50: “as well as about aggression and dominance [23].” The article focusses on aggression, not on dominance. Omit “dominance”.

Response: Thank you, we have omitted it.

Line 52: faces or faeces?

Response: Thank you very much, we have changed it to “faeces”.

Once again, thank you for your all expert comments. We hope that you will be satisfied with the current version of the manuscript.

Reviewer 3 Report

Review of Interspecific chemosensory communication of emotions

General comments

The fear condition to which the subjects are subjected contains a strong element of physical activity which, in and of itself, is bound to increase sweat production. It is very likely that the condition is also psychologically stressful for the subjects but to what extent is unclear. The authors claim it is (line 90.91: ‘we expected that the horses might experience fear’ and line 92: ‘These events are commonly known to be particularly stressful’) but have no physiological evidence that it is. Similarly, it is also unclear (and unknown) to what extent the condition elicits fear in the subjects. In other words, physically stressful, yes; psychologically stressful, maybe; fearful, maybe.

It is unclear whether the discrimination between fear and non-fear odor is based on quantitative or qualitative differences. In lines 172-175 it is stated that: ‘fear odour was perceived as more intense in 64% of trials, and the non-fear odour in only 16.1% of trials and thus in a significantly higher proportion.’ In line 177-180 the authors claim that ‘humans, as a group, were able to correctly assign whether horse odour samples were collected under a fear- or a non-fear condition, respectively. Furthermore, they perceived the body odour of horses collected under the fear condition as more intense compared to the non-fear condition.’ Finally, in line 205-209: ‘Humans not only correctly assigned the emotional state to the horse odour samples but also perceived the body odour of horses collected under the fear condition as more intense compared to the non-fear condition and thus displayed the basic ability to discriminate between the odour samples (based on intensity).’

Taken together, it appears that discrimination is primarily based on odor intensity rather than odor quality and that qualitative differences were only detected by a few humans, a result that I think should be added to the conclusion in the simple summary (line 17-18), the abstract (line 30-31), and to the conclusions (line 227-228).

In other words, maybe the odor raters were just able to identify more sweaty horses.

Specific comments:

Line 93 & 103: ear position, which one? Behavior, which kind of behavior?

Line 96: ‘cleaned’. How was it done? With water (i.e. the odor diluted) or with some kind of detergent (i.e. an odor added)?

I recommend acceptance of the manuscript with major revisions. As a minimum, the authors must discuss these uncertainties of their study.

Author Response

Dear Reviewer,

Thank you very much for your expert review. We really appreciate your detailed comments and all the hints you gave us – following your suggestions, we incorporated many changes to our manuscript. Please find our responses below.

The fear condition to which the subjects are subjected contains a strong element of physical activity which, in and of itself, is bound to increase sweat production.

Response: We carefully considered the reviewer’s suggestion and now address this concern. The corresponding sentences now read as follows: Thirdly, the fear condition contained a strong element of physical activity which is bound to increase sweat production. As a result, an open question remains whether humans could simply distinguish between little versus much sweat and between high-intensity versus low-intensity or were able to recognise hoses fear and non-fear emotions. While future research in this field should control for the level of physical activity in both conditions, it should be noted that experiencing fear per se leads to increased sweating [67]. Thus, we assume that the horses could sweat more not only because of increased physical activity but also experiencing more fear.

It is very likely that the condition is also psychologically stressful for the subjects but to what extent is unclear. The authors claim it is (line 90.91: ‘we expected that the horses might experience fear’ and line 92: ‘These events are commonly known to be particularly stressful’) but have no physiological evidence that it is. Similarly, it is also unclear (and unknown) to what extent the condition elicits fear in the subjects. In other words, physically stressful, yes; psychologically stressful, maybe; fearful, maybe.

Response: We took up the reviewer’s suggestion and now address this concern. The corresponding sentences now read as follows: The present study is not free from limitations. Firstly, the horses’ reactions in the fear condition were only validated via behavioural measures. Thus, it may be insufficient to conclude whether this condition has actually generated fear and, if so, to what extent. Future research should include other options of verifying the emotional reaction of horses, such as measuring cortisol levels in horses blood.

It is unclear whether the discrimination between fear and non-fear odor is based on quantitative or qualitative differences. In lines 172-175 it is stated that: ‘fear odour was perceived as more intense in 64% of trials, and the non-fear odour in only 16.1% of trials and thus in a significantly higher proportion.’ In line 177-180 the authors claim that ‘humans, as a group, were able to correctly assign whether horse odour samples were collected under a fear- or a non-fear condition, respectively. Furthermore, they perceived the body odour of horses collected under the fear condition as more intense compared to the non-fear condition.’ Finally, in line 205-209: ‘Humans not only correctly assigned the emotional state to the horse odour samples but also perceived the body odour of horses collected under the fear condition as more intense compared to the non-fear condition and thus displayed the basic ability to discriminate between the odour samples (based on intensity).’

Taken together, it appears that discrimination is primarily based on odor intensity rather than odor quality and that qualitative differences were only detected by a few humans, a result that I think should be added to the conclusion in the simple summary (line 17-18), the abstract (line 30-31), and to the conclusions (line 227-228). In other words, maybe the odor raters were just able to identify more sweaty horses.

Response: We took up the reviewer’s suggestion and now address this concern. The corresponding sentences now read as follows: Thirdly, the fear condition contained a strong element of physical activity which is bound to increase sweat production. As a result, an open question remains whether humans could simply distinguish between little versus much sweat and between high-intensity versus low-intensity or were able to recognise hoses fear and non-fear emotions. While future research in this field should control for the level of physical activity in both conditions, it should be noted that experiencing fear per se leads to increased sweating [67]. Thus, we assume that the horses could sweat more not only because of increased physical activity but also experiencing more fear.

Furthermore, following your suggestion, we have critically reviewed the simple summary, abstract, and conclusion. The corresponding sentences now read as follows: However, these results should be addressed with caution in light of the study's limitations and only viewed as exploratory for future studies (…) To conclude: the results of the present study suggest that olfactory cues might contribute to the recognition of horse emotions by humans. An open question remains whether humans could simply distinguish between little versus much sweat and between high-intensity versus low-intensity or were able to recognise hoses fear and non-fear emotions. Future studies with carefully elaborated methodology should explore this issue further.

Eventually, we decided to change the manuscript’s title into: “A preliminary investigation of interspecific chemosensory communication of emotions: humans (Homo sapiens) can recognise fear- and non-fear body odour from horses (Equus ferus caballus)”.      

Line 93 & 103: ear position, which one? Behavior, which kind of behavior?

Response: Thank you very much for pointing out these inaccuracies; we have corrected them. The corresponding sentence now reads as follows: Both ears position, head position, and body tension of the horses during this training confirmed our expectation.

Line 96: ‘cleaned’. How was it done? With water (i.e. the odor diluted) or with some kind of detergent (i.e. an odor added)?

Response: We took up the reviewer’s suggestion and now added more details on horse cleaning. The corresponding sentence now reads as follows: In the second step, after a minimum of 20 minutes, the horses were cleaned with fresh straw and brushed, and new pieces of fleece material of the same dimensions, together with new saddle-pads and lunging belts were put on them.

We also referred to this aspect in the discussion. The corresponding sentences now read as follow: Eventually, several methodological issues should be considered. Relatively short time-lag of 20 min. between the fear and non-fear situation, limited possibilities of cleaning the horses between both conditions, and not randomised order of treatments are weaknesses of this study. This being said, it must be noted that the study was conducted in natural, non-laboratory conditions and changing these aspects was not practically feasible.

Thank you very much for all your kind remarks. We have responded to all your remarks.

Round 2

Reviewer 1 Report

Although I rejected this paper in the first round, the authors were more conservative and careful with their data in this new version. I continue to believe that the raters confused the fear samples with those more intense. However, my feeling is not science, and thus these results need to be submitted to the scientific community after some minor adjustments.

The title should be changed again.

Replace “humans (Homo sapiens) can recognise fear- and non-fear body 4 odour from horses (Equus ferus caballus)” with “can humans (Homo sapiens) recognize fear- and non-fear body odor from horses (Equus ferus caballus)?

Line 52: replace “animals” with “non-human animals”

Lines 190-2: Limitations are treated in detail further on, then remove this sentence. Here could make confusing the reader.

Lines 225-6: replace “situations” with “social contexts”

Line 90: add the following sentence after “…showing ability of purely olfactory recognition of human emotions.”: Moreover, human body odours were shown to induce systematic sympathetic and parasympathetic changes in horses, revealing that the human chemosignals are able to induce similar emotional status in horses (Semin et al., 2019), as also demonstrated in dogs (D’Aniello et al., 2018).

Additional Reference:

Semin, G. R., Scandurra, A., Baragli, P., Lanatà, A., & D’Aniello, B. (2019). Inter-and intra-species communication of emotion: chemosignals as the neglected medium. Animals, 9(11), 887.

I hope my suggestions were helpful

Author Response

Dear Reviewer,

Thank you very much for expressing your positive opinion about the current version of our manuscript. We were more than glad to hear that you changed your mind and accepted the study for publication. Thank you for your detailed comments and suggestions; following them we incorporated many changes to the manuscript. Please find our responses below.

The title should be changed again. Replace “humans (Homo sapiens) can recognise fear- and non-fear body 4 odour from horses (Equus ferus caballus)” with “can humans (Homo sapiens) recognize fear- and non-fear body odor from horses (Equus ferus caballus)?

Response: We took up the reviewer’s suggestions and changed the title accordingly. 

Line 52: replace “animals” with “non-human animals”

Response: Thank you very much, we have replaced it.

Lines 190-2: Limitations are treated in detail further on, then remove this sentence. Here could make confusing the reader.

Response: We took up the reviewer’s suggestion and removed this sentence.

Lines 225-6: replace “situations” with “social contexts”

Response: Thank you, we have replaced it.

Line 90: add the following sentence after “…showing ability of purely olfactory recognition of human emotions.”: Moreover, human body odours were shown to induce systematic sympathetic and parasympathetic changes in horses, revealing that the human chemosignals are able to induce similar emotional status in horses (Semin et al., 2019), as also demonstrated in dogs (D’Aniello et al., 2018).

Response: We carefully considered the reviewer’s suggestion and followed the given advice. The corresponding sentence now reads as follows: Moreover, human body odours were shown to induce systematic sympathetic and parasympathetic changes in horses, revealing that the human chemosignals are able to induce similar emotional status in horses [43,45]

Please, notice that we added here Lanata and colleagues (2019), who conducted the study on horses, together with Semin and colleagues (2019), the authors of a valuable overview cited by the reviewer. We did not refer here to D’Aniello et al. (2018) because his work was already mentioned in this paragraph: D’Aniello et al. [41] reported that dogs are able to recognise human emotions from body odour as they displayed behaviours indicating stress only when being presented with human odour of fear (lines 84-86).

Additional Reference:Semin, G. R., Scandurra, A., Baragli, P., Lanatà, A., & D’Aniello, B. (2019). Inter-and intra-species communication of emotion: chemosignals as the neglected medium. Animals, 9(11), 887.

Response: Thank you very much for indicating this interesting paper; we have referred to it.

Thank you very much for all your remarks. We have addressed them all. Thanks to all your suggestions and comments, we have improved our manuscript. We hope you will be satisfied with the current version of the manuscript.

Reviewer 2 Report

Dear authors, thank you for your new version. I like the more critical and cautious interpretations. I think the article can inspire other researchers to dig deeper into this interesting research question. 

Author Response

Dear Reviewer,

Thank you very much for expressing your positive opinion about the current version of our manuscript. We also hope that the article can inspire new research in this field after all the changes incorporated to our manuscript according to your suggestions. Once again, we would like to thank you for guiding us with your expert comments and suggestions. 

Reviewer 3 Report

The auyhors have taken my comments and questions plus those of the other reviewer into consideration. The result, however, is a manuscript that contains so many caveats that I do not think it is worthy of being published. The fact that the title has been changed to ‘A preliminary investigation’, and the text changed to ‘an open question’ (line 18), ‘olfaction might contribute’ (line 21), ‘these results should be addressed with caution’ (line 22) are just a few examples of some major weaknesses of the study.

I therefore recommend rejection of the manuscript.

Author Response

Dear Reviewer,

Thank you very much for your expert review. We understand your concerns regarding the study itself. Also, other reviewers shared your criticism in the first round of reviews; one recommended rejection of the paper. In response, we underlined our awareness of the weaknesses of the study and willingness to discuss them openly. As a result, both reviewers spoke positively about the paper in the second round. Please, let us emphasize that with all the weaknesses, this is the first study to indicate a fascinating notice that humans might be able to recognise the emotions of different species. Therefore, it can set directions for future research with a carefully planned methodology. We believe that, by discussing all the caveats and open questions openly, we present promising results of an experiment conducted in natural conditions. Thank you once again for sharing with us your point of view.